

# Assessing the key concerns in snow storage: A case study for China

Xing Wang[1,2], Feiteng Wang[1], Jiawen Ren[1,2], Dahe Qin[1,2], Huilin Li[1]

[1]State Key Laboratory of Cryospheric Science, Northwest Institute of Eco-Environment and Resources,

Chinese Academy of Sciences, Lanzhou 730000, China

[2]University of Chinese Academy of Sciences, Beijing 100049, China

*Correspondence*: Feiteng Wang (wangfeiteng@lzb.ac.cn)

**Abstract.** Snow security plays a crucial role in ensuring the success of winter sports events and

supporting the ski industry. One effective approach to enhancing snow reliability is through snow

storage. Despite its potential benefits, the investigation of snow storage in China has been limited. To

address this gap, we implemented snow storage covered with geotextile at two venues, namely the Big

Air Shougang (BAS) in Beijing for the 2022 Winter Olympic Games and the National Biathlon Center

(NBC) in Chongli for the 2022 Winter Paralympic Games, in response to extreme meteorological

conditions. In order to assess the key concerns associated with snow storage: ablation (the process of

snow loss) and snow quality, we introduced a fine snow pile monitoring system and utilized the

SNOWPACK model. Our observations revealed that, by February 18, the snow pile at the BAS had lost

158.6 m$^3$ of snow (equivalent to 6.7% of the initial volume). Subsequently, the accelerated ablation was

influenced by both meteorological conditions and the presence of a thin geotextile layer. Between

January 16 and April 15, the snow pile at BAS experienced a total loss of 1242.9 m$^3$ of snow

(corresponding to 52.7% of the original volume). With regard to snow quality, there were no significant

variations modeled at the study sites, except for the upper part of the snow piles. It was noted that the

evaporation of wet geotextile contributed to slowing down the ablation process. Consequently, we

discourage the use of impermeable coverage schemes for snow storage. Instead, the thickness of the

cover played a vital role in preserving the snow pile. At Beijing and Chongli, the 0.7 m and 0.4 m thick

cover layers, respectively, were found to protect approximately half of the snow height over the

summer season. Importantly, the snow quality evolution was consistent across different cover

thicknesses. The findings of our study have implications for the ski industry in China, as they provide

valuable insights into snow storage techniques and their impact on snow reliability.



## 1   Introduction

Snow storage is the conservation of snow covered with thermal insulation materials over a period, following the principles of low environmental impact and cost-effectiveness. Winter sports like skiing are highly sensitive to climate change, and the effects of global warming and uncertain winter snowfall can significantly impact winter sports events and the ski industry. In this context, snow storage is considered a more reliable approach than snowmaking for ensuring snow security, as it requires

specific weather conditions such as low air temperature, relative humidity, and wind speed (Olefs et al., 2010; Spandre et al., 2017). Implementing snow storage not only ensures smooth competition but also portrays a positive image in terms of snow security and the professionalism of event hosts (Grünewald et al., 2018). Snow storage has been successfully used in various major winter sports events in the past to tackle unpredictable meteorological conditions, including events such as the cross-country in Davos,

Altay, and Vuokatti, the biathlon world cup in Östersund, the ski jumping world cup in Titisee-Neustadt, and the 2014 Winter Olympics in Sochi (Grünewald et al., 2018; Lintzén and Knutsson, 2018; Pestereva, 2014). Additionally, ski resorts have utilized snow storage in Scandinavia, the Alps, and Canada for several years. In past warm ski seasons, the ski season length, skiable terrain, snow quality, and snowmaking days were reported to decrease in Canada (Rutty et al., 2017). As climate change

continues, snow reliability is expected to decrease, particularly for low and mid-altitude ski resorts (Demiroglu et al., 2016; Willibald et al., 2021).

Snowmaking and snow storage have become essential elements of modern snow management in ski resorts (Steiger and Scott, 2020; Weiss et al., 2019). Snow storage serves as a complementary solution to address the limitations of snowmaking due to meteorological conditions and resource constraints. By

using snow storage, ski resorts can ensure an exact opening date for the ski season and even extend the season when snowmaking is not feasible or efficient. For instance, in Piteå, nearly 2500 m$^3$ of snow covered with bark was stored to guarantee an early start for the ski track opening (Lintzén, 2012). Grünewald et al. (2018) demonstrated the use of a snow pile covered with sawdust for cross-country skiing in Switzerland. Weiss et al. (2019) explored the feasibility of snow storage in the northern

United States. An approach that involves collecting natural snow from winter and spring snowfall events and storing it for ski track construction could help reduce electricity and water consumption for



snowmaking. Geotextile has been shown to decrease the total ablation of snow and ice by around 60% in an Austrian glacier ski resort (Olefs and Fischer, 2008). Furthermore, there is increasing interest in using stored natural snow for summer building refrigeration (Hamada et al., 2012; Kumar et al., 2016;

Skogsberg and Nordell, 2001).

Snow storage can be carried out using four different methods: indoor storage, on-ground storage, open pit storage, and underground storage (Skogsberg and Nordell, 2006). These approaches aim to alter the snow surface energy balance and slow down the melting rate of snow piles. Indoor and open pit storage methods are particularly useful for extracting cold energy from snow, making them suitable for

building refrigeration purposes (Hamada et al., 2007; Moe, 2018). However, storing snow in open pits and underground is limited by site constraints, and indoor storage is not practical for large-scale snow storage. As a result, the most commonly employed method in the ski industry is storing snow on the ground with thermal insulation materials, providing flexibility in implementation. For instance, Ter-Zakaryan et al. (2019) stored snow on a ski track at the end of the ski season. At glacier ski resorts, the

approach involves placing insulation materials directly on top of the natural snow when it reaches its maximum height and removing the materials before the first persistent snowfall occurs in late summer or early autumn (Olefs and Fischer, 2008). On the other hand, glacier-free ski resorts use the method of collecting natural and/or artificial snow, grooming it, and storing it with thicker insulation materials (Weiss et al., 2019).

Detailed research on snow storage for the ski industry, especially in the Asian region, is still limited. Many aspects of snow storage remain unknown, such as snow quality, localized storage schemes, the evolution of snow piles, and the factors influencing them. To address this knowledge gap and ensure snow safety during the Beijing 2022 Winter Olympic and Paralympic Games under extreme meteorological conditions, we performed snow storage. Utilizing a terrestrial laser scanner, we

obtained high-resolution repeated geometrical data of the snow piles. Additionally, we conducted continuous monitoring of the thermal conditions of the snow piles, a detail not frequently found in published literature. This comprehensive analysis allows for a thorough examination and evaluation of the evolution of snow piles and the factors affecting them. We also explored snow storage over the summer season in Beijing and Chongli, which are regions with a high concentration of Chinese ski

areas (An et al., 2019). In summary, the main objectives of this study are: (i) to investigate the

evolution of snow piles and the factors influencing them, (ii) to assess the performance of the

SNOWPACK model in China, particularly in mid-latitude and low-altitude regions, and (iii) to identify

significant considerations in snow storage. By accomplishing these aims, we aim to provide a

comprehensive technical procedure and knowledge of snow storage and its effects, given the rapid

growth of the Chinese ski industry.

## 2    Study sites and snow storage for Beijing 2022 Winter Olympic and Paralympic Games

We implemented snow storage at two venues, namely the Big Air Shougang (BAS) and the National

Biathlon Center (NBC) to ensure a smooth competition process. The BAS, located in the central urban

area of Beijing, served as the Winter Olympic skiing events venue. The storage site (39.908°N,

116.147°E, 75.8 m a.s.l.) is 0.36 km away in south from the track of the BAS (Figure 1a and 1b), lying

in cleared and flat soil ground. Such a short distance facilitates the transportation of stored snow. The

surrounding buildings and fences provided additional wind protection to the snow pile. Snow

production for storage began on December 26, 2021, and continued until January 11, 2022, utilizing

two snowmakers. The shape of the snow pile was adjusted by moving the snowmakers. On January 3,

2022, we started installing an automatic meteorological station, thermal sensors, and gathering the

initial snow density data. Taking into account various factors, including meteorological conditions,

availability of materials, and construction considerations, a 0.004 m thin reflective layer (geotextile)

was used to cover the snow pile on January 21, 2022. The geotextiles were securely fastened and

connected using white ropes. Finally, in late April, the BAS staff removed the snow pile.

As for the NBC, it is situated in the Chongli district, approximately 52.63 km northeast of Zhangjiakou.

At this location, the second snow pile (40.912°N, 115.469°E, 1639.5 m a.s.l.) was established adjacent

to para tracks by adjusting snowmakers and covering the pile with a 0.004 m thin geotextile on January

20, 2022 (Figure 1c and 1d). Rocks wrapped in geotextile and white ropes were used to secure the

geotextile layer. The site at NBC also had cleared soil ground but was harder compared to the BAS site.

Additionally, it was surrounded by mountains on the east and west sides. Snow production for storage

took place between January 12 and January 16, 2022, benefiting from lower air temperatures. A



monitoring system for snow storage, similar to the one used for the BAS pile, was installed. In early

March, the stored snow was removed using groomers to prepare the track for para cross-country skiing

and para biathlon.

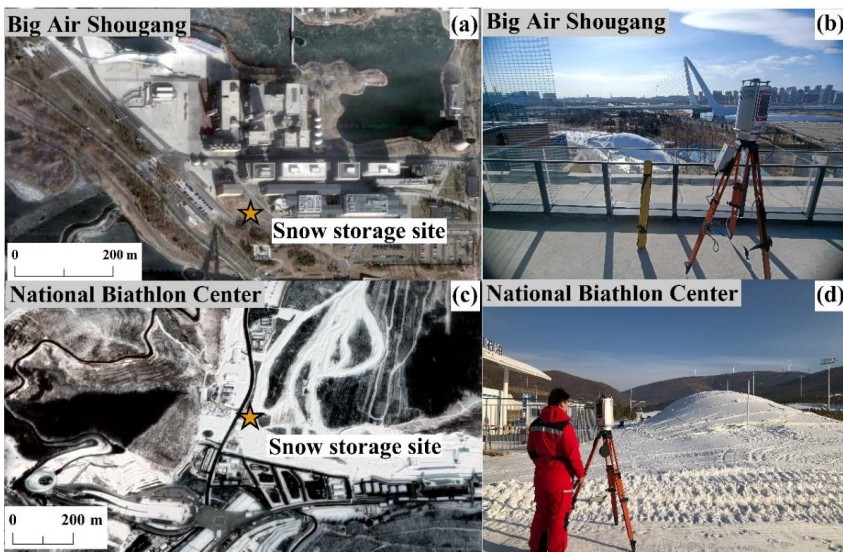


**Figure 1:** Snow storage at the Big Air Shougang (BAS) and the National Biathlon Center (NBC), 2022. Snow storage sites at the BAS (a) and the NBC (c) (satellite images from ©Google earth). The snow pile before covered on January 16, 2022 at the BAS (b); the snow pile covered with geotextile on January 20, 2022 at the NBC (d) (photos provided by Xing Wang).

## 3    Data and methods

### 3.1    Meteorological data

The main meteorological conditions surrounding the snow piles were recorded using two automatic meteorological stations. These stations were installed adjacent to the snow piles on January 10 and January 20, 2022, at the BAS and the NBC, respectively. Due to venue management restrictions, the

sensors' height was maintained at 2–3 m above the ground. The parameters were measured at a 15-minute resolution included air temperature, relative humidity, incoming and outgoing shortwave radiation and longwave radiation, wind speed, maximum wind speed, and wind direction. Precipitation data were obtained from the Mentougou national meteorological station (39.888°N, 116.156°E, 85.5 m a.s.l.), located 2.35 km away from the BAS storage site, and the Chongli national meteorological





station (40.954°N, 115.269°E, 1239.8 m a.s.l.), situated 17.44 km away from the NBC site. Additionally, long-term meteorological data from these national stations for the years 2019–2022 were acquired from the National Meteorological Science Data Center to assess snowmaking conditions and snow storage over the summer season. Since incoming shortwave radiation data were unavailable for the Mentougou and Chongli national meteorological stations, this information was provided by the

Daxing national meteorological station (39.718°N, 116.354°E, 37.5 m a.s.l.) for Beijing and the Zhangbei national meteorological station (41.150°N, 114.700°E, 1393.3 m a.s.l.) for Chongli.

## 3.2   Snow pile data

A monitoring system for the snow piles was established, and the information regarding monitoring parameters, sensors, and devices can be found in Table 1. The repeated geometrical data of the snow

piles were obtained using a *Riegl* VZ®-6000 Class 3B terrestrial laser scanner (TLS). Four temporal terrestrial laser scanning data were collected at the BAS on January 16 (TLS1), February 18 (TLS2), March 9 (TLS3), and April 15 (TLS4), 2022, respectively. As for the NBC, one terrestrial laser scanning survey was performed on January 20, 2022, and no additional data was available for subsequent times. To continuously monitor the thermal conditions of the snow piles, a series of sensors

were installed. In January 2022, the snow density was measured 3 m below the top surface using an ice core drilling with extension rods at both the BAS and the NBC. Further details can be found in the supplementary material.

**Table 1:** Snow piles monitoring parameters, sensors and devices information at the BAS and the NBC.

| Parameters | Sensors or devices | Accuracy |
|---|---|---|
| Snow pile spatial information | Riegl VZ®-6000 | ± 15 mm |
| External geotextile surface temperature | Infrared radiometer sensor SI-111-SS | ± 0.2 °C for -10 to 65 °C ± 0.5 °C for -40 to 70 °C |
| Snow-geotextile interface temperature | Thermistor sensor PT100 | ± 0.15 °C for -50 to 200 °C |
| Snow-geotextile interface heat flux | Foil heat flux sensor FHF02SC | -- |
| Internal snow temperature | Thermistor sensor temperature chain | ± 0.02 °C for -30 to 30 °C |
| Density | Electronic scale | ± 0.01 g |
| | Ice core drilling | -- |

## 3.3   TLS data processing

Geographic analysis cannot directly utilize the raw point clouds exported from the TLS. Therefore, we

employed the software RiSCAN PRO® v 2.14 (RIEGL Laser Measurement Systems, 2022) for post-processing, with most operations automated. Initially, we removed noise points with low reflectance and/or high deviation, as well as points isolated from all scan stations, to eliminate their impact on subsequent registrations. At this stage, the point clouds from each scan station were treated as

independent entities. The second step involved automatic registration to combine these individual point clouds. Subsequently, a fine registration was achieved through multi-station adjustment based on plane patches derived from the point clouds. In the third step, the registered point clouds were transformed into a global coordinate system (UTM 50N) using tie points created from the ground control points. Lastly, we exported the processed point clouds, filtered to 0.01 m in XYZ directions (octree mode), as

the las format. These exported point clouds were then utilized to generate 0.05 m resolution DSMs using ESRI ArcMap 10.6. The geographic analysis and processing of these DSMs were conducted in ESRI ArcMap 10.6 and PyCharm IDE 2021.

## 3.4   Snow model

The SNOWPACK model is a one-dimensional physical model that utilizes a finite-element method to

provide a detailed description of mass and energy exchange between snow, atmosphere, and soil (Bartelt and Lehning, 2002; Luetschg et al., 2008). It has been widely used to study snow evolution (Horton and Haegeli, 2022; Keenan et al., 2021; Wever et al., 2015). For our study, we employed the SNOWPACK model to simulate the top position evolution of the snow piles where the geotextile-snow interfaces were fitted with sensors and unaffected by geotextile overlaps at both the BAS and the NBC.

Additionally, the model was utilized to simulate snow storage over the summer season at the top of the snow piles.

In the SNOWPACK model, the meteorological forcing data were obtained from nearby automatic meteorological stations at the BAS and the NBC. The initial densities and temperatures of the snow layers were based on measurements, while the initial snow grain radius and liquid water content were

set to 1 mm and 3%, respectively, based on experiences and previous studies (Grünewald et al., 2018). The SNOWPACK model allows for the inclusion of other materials as the topmost layer, with specified mechanical and thermal properties (Grünewald et al., 2018; Luetschg et al., 2008; Olefs and Lehning,

2010). All other materials are represented as a three-component porous matrix (solid, void, and water).

Valuable information on snow storage simulation has been provided by Rinderer (2009) and Grünewald

et al. (2018). For the 0.004 m thin geotextile layer, the initial density and thermal conductivity were set

at 111.1 kg m$^{-3}$ and 0.06 W m$^{-1}$ K$^{-1}$, respectively, while other properties were obtained from the

literature (Rinderer, 2009). In the simulation of snow storage over the summer season, an all-sky

algorithm was applied to generate incoming longwave radiation, satisfying the meteorological

requirements in the SNOWPACK model. The properties of the sawdust and straw chips layers in the

model were assumed to be the same as described in previous studies (Grünewald et al., 2018; Rinderer,

2009), except for the thermal properties and densities, which were taken from a previous study (Wang

et al., 2021).

## 4    Results

### 4.1    Meteorological conditions and snow quality

Figure 2 presents the diurnal meteorological parameters from the two nearby automatic meteorological

stations during specific periods: January 11 to April 27, 2022, at the BAS, and January 21 to March 13,

2022, at the NBC. A significant difference in mean air temperatures was observed between January 21

and March 13 at the two snow storage sites, with 1.5 °C at the BAS and –11.1 °C at the NBC. Trends of

increasing and then decreasing air temperatures and relative humidity were noted during the Winter

Olympic competition days (WOcd) at the BAS (February 7 to 15) and the NBC (February 5 to 19)

(Figure 2a and 2c). The air temperature gradually rose between the WOcd and the Paralympic

competition days (WO-WPcd) at the NBC (February 20 to March 4), which was not favorable for

snowmaking. In contrast, the variations in relative humidity were conducive to snowmaking during the

same period (Figure 2c). Overall, the wet bulb temperature remained well below the threshold for

snowmaking (Sect. 5.1), indicating the availability of artificial snow before the Paralympic competition

days (WPcd) at the NBC. The mean air temperature, relative humidity, incoming shortwave radiation,

and longwave radiation for the entire record at the BAS were 5.9 °C, 40.6%, 160.8 W m$^{-2}$, and 287.2 W

m$^{-2}$, respectively, while those at the NBC were –11.1 °C, 56.7%, 157.3 W m$^{-2}$, and 200.3 W m$^{-2}$,

respectively. The mean wind speed was 0.5 m s$^{-1}$ at the BAS and 1 m s$^{-1}$ at the NBC. The predominant

wind directions were opposite at the BAS and the NBC (Figure 2e and 2f), possibly due to local

topography. Precipitation was scarce, resulting in reduced track maintenance needs, with cumulative

precipitations of 36.7 mm at the BAS and 14.2 mm at the NBC.

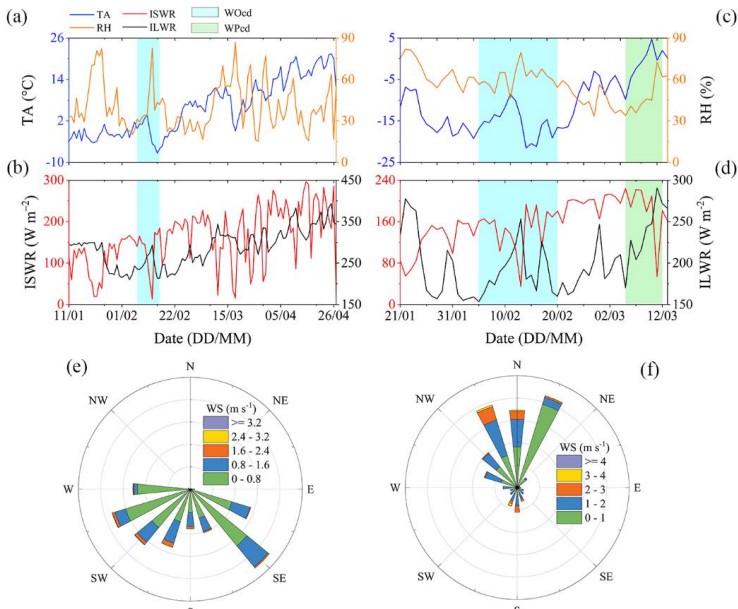

**Figure 2:** Diurnal meteorological conditions at the BAS (a, b, e) (January-April) and the NBC (c, d, f) (January-
March), 2022. (a, c) The blue and orange lines represent air temperature (TA) and relative humidity (RH),
respectively. (b, d) The red and black lines indicate incoming shortwave radiation (ISWR) and incoming longwave
radiation (ILWR), respectively. (e, f) Wind speed (WS) and wind direction (WD). The vertical light cyan shading
signifies the Winter Olympic competition days (WOcd) at the BAS (7 to 15 February) and the NBC (5 to 19
February). The vertical light green shading signifies the Winter Paralympic competition days (WPcd) at the NBC
(5 to 13 March). The BAS is not a Winter Paralympic venue.

On January 16, three groups of snow cores were extracted and weighed from 3 m below the top surface

of the snow pile at the BAS. The mean density of the snow cores (1 m below the top surface) was 459.5

kg m$^{-3}$ at the BAS. The mean density increased with depth, but the values at 2 and 3 m were close, at

533.5 and 538.7 kg m$^{-3}$, respectively. Similarly, at the NBC, the mean densities at 1, 2, and 3 m below

the top surface were 478.1, 515.3, and 529.0 kg m$^{-3}$, respectively. When snow falls on the ground,

continuous changes in grain size, shape, and cohesion affect the physical properties of the snow. On

January 7, February 7, February 20, and March 11, an increase in grain size over time with low

cohesion on the top surface was observed at the BAS. On February 20, an "ice layer" formed by



freezing top surface meltwater existed 0.1–0.2 m below the top surface. This layer is characterized by

low permeability and porosity (Colbeck, 1982), meaning less heat from the surface is transferred by the

meltwater to the snow below the "ice layer". The pores above the "ice layer" were gradually filled with

liquid water, and horizontal flow might occur (Colbeck, 1972).

## 4.2 Digital surface models (DSMs) and mass changes derived from TLS data

Four TLS surveys were conducted to generate DSMs at the BAS (Figure 3e–3h) which were then used

for hillshade rendering (Figure 3a–3d) and mass changes analysis (Figure 3i–3k). However, at the NBC,

only one TLS survey was available, so no change analysis was provided. The absolute volume of the

snow pile required a bare ground elevation model, which was unavailable for both the BAS and the

NBC. Therefore, it was approximated through interpolation based on the nearby ground. The initial

snow pile volume at the BAS was 2356.6 m³. Due to the snow storage site's limitations, the shape of

the snow pile was long and narrow, resulting in a high upper surface area-to-volume ratio of 0.63. The

snow pile edges with low snow depths in a large area may also explain this high value. Additionally,

the snow pile surface was noticeably uneven over time (Figure 3a–3d), caused by the nonuniform

melting between the geotextile overlaps and other parts. Beyond the WOcd, a TLS survey was

performed on February 18 at the BAS, revealing a slight volume change of 158.6 m³ snow loss between

January 16 (TLS1) and February 18 (TLS2). From the second (February 18) to the third survey (March

9), 304.2 m³ of snow was lost, and a significant snow loss of 780.1 m³ occurred from March 9 (TLS3)

to April 15 (TLS4). This considerable snow loss after the WOcd was expected, and the accelerated

melting was mainly influenced by meteorological conditions. The thin geotextile layer could not

provide adequate protection as air temperature rose (Olefs and Lehning, 2010; Rinderer, 2009).

Furthermore, fouling on the external geotextile surface decreased shortwave radiation reflectivity. The

snow pile showed an evident tendency of increasing ablation on the long sides, particularly the

southwest side. Three main factors contributed to this tendency: coverage, wind, and shortwave

radiation. An air layer existed between the snow surface and geotextile on the long sides due to the

geometry, while the overlap facilitated air circulation between the internal and external geotextile

layers, promoting heat exchange without thermal resistance from the geotextile layer. Wind erosion

also contributed to increased snow loss, with the southwest side being windward of the snow pile



(Figure 2e), while the northeast side was leeward, leading to convective heat exchange differences. In the morning, buildings obscured the direct shortwave radiation reaching the northeast side, but it remained unimpeded when the sun was in the south, southwest, and west orientations (Figure 1a).

Densification and ablation affected the snow height variations. The high initial density reduced settlement, resulting in a 6.0% increased density on April 15 (Sect. 4.4). Consequently, melting accounted for 1101.5 m³ of snow loss, while densification contributed to 141.4 m³. The DSM and hillshade rendering at the NBC are shown in Figure 4a and 4b, respectively, with an initial snow pile volume of 2995.2 m³, which was 638.6 m³ larger than that of the BAS.

There are noticeable negative values in snow height and snow height variations at the edges of the snow pile due to different reasons. In Figure 3e–3h, this occurs because the bare ground interpolation is higher than the actual ground, as evident from the expansion of the negative area over time. In Figure 3i, the negative values result from the presence of natural snow. In Figure 3j and 3k, irregular stacks of geotextiles appear to be the primary reason. On 18 February, we observed that natural snow on the long

sides accumulated at the edges of the snow pile due to gravity, and at that time, the artificial snow beneath the geotextile was slightly lost. Instead, most of the artificial snow at the edges disappeared, and wind caused the geotextiles to form irregular stacks on 9 March and 15 April. Since the negative areas have low snow depths, they have a minor impact on volume analysis and are treated as zero.



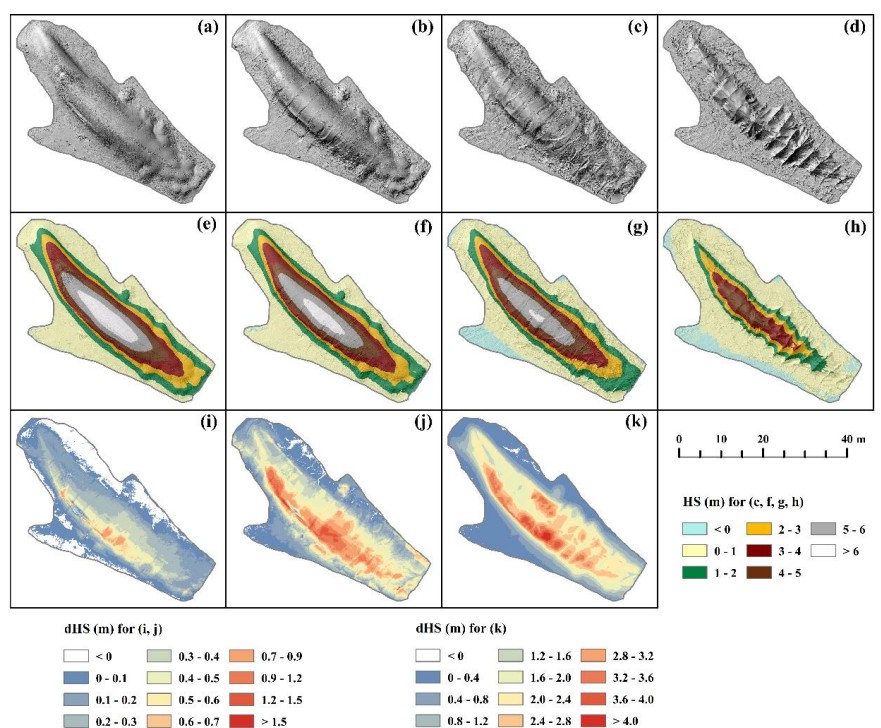

**Figure 3:** The BAS snow pile. Hillshade rendering based on the TLS-derived DSMs on January 16 (TLS1) (a), February 18 (TLS2) (b), March 9 (TLS3) (c), and April 15 (TLS4) (d), 2022. Spatial distribution of snow height (HS) referenced by the interpolated ground for TLS1 (e), TLS2 (f), TLS3 (g), TLS4 (h). Spatial distribution of snow height variations (dHS) for TLS1-TLS2 (i), TLS1-TLS3 (j), and TLS1-TLS4 (k).

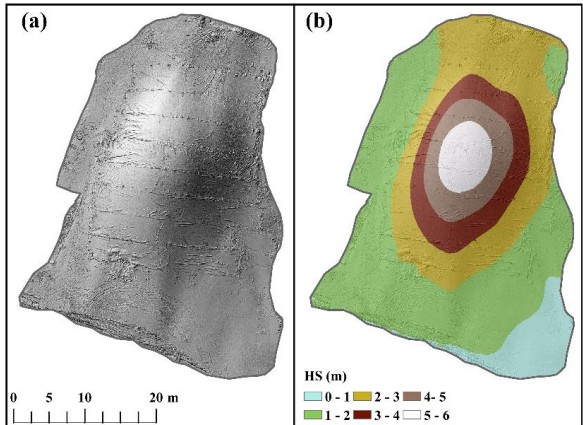

**Figure 4:** The NBC snow pile. Hillshade rendering based on the TLS-derived DSM on January 20, 2022 (a). Spatial distribution of snow height (HS) referenced by the interpolated ground (b).





### 4.3    Measured snow pile thermal regime

The external geotextile surface temperature on the southwest side was found to be more sensitive to meteorological conditions than the northeast side at the BAS (Figure 5a), likely due to factors such as

wind, radiation, and the presence of an air layer. The slight difference in external geotextile surface temperature on the snow pile's long sides before January 25 at the BAS can be explained by two reasons. First, the snow pile was not immediately covered until January 21, so the data represented the snow surface temperature on the long sides between January 17 and 20. Second, a snowfall event occurred between January 21 and 24, leading to the geotextile surface being covered with natural snow

and subsequently re-recording the snow surface temperature. Phase transitions on the snow surface can moderate temperature variations. In hourly temperature data from January 17 to April 24, the mean peak temperatures on both long sides occurred together at 14:00, with 0.9 °C on the northeast side and 1.1 °C on the southwest side at the BAS, influenced by incoming shortwave radiation, air temperature, and terrain shadowing (Sect. 4.2). The underlying snow surface, longwave radiation, and evaporation

strongly cooled the external geotextile surface. The external geotextile surface temperature was lower than the air temperature at both the BAS and the NBC. The mean external geotextile surface temperatures on the two sides were nearly equal (Figure 5b), with –19.7 °C on the northwest side and –19.4 °C on the short southwest side at the NBC. The maximum hourly temperature on the northwest side occurred at 14:30 with –11.9 °C, half an hour later than the short southwest side with –10.8 °C.

This difference was also due to the solar incidence angle on the snow pile.

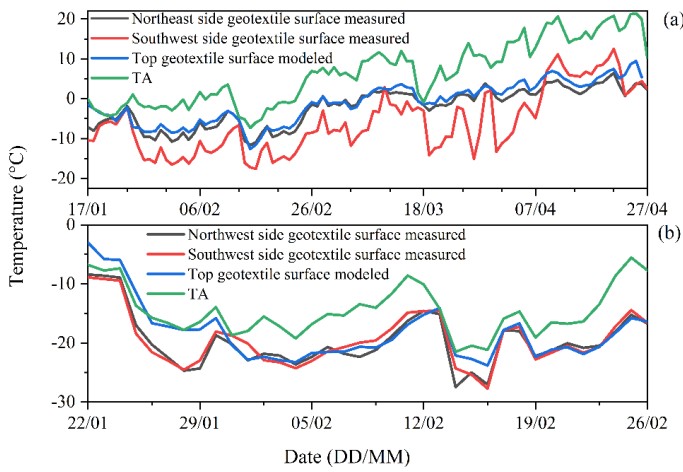



**Figure 5:** Diurnal measured external geotextile surface temperatures on snow piles sides and modeled external geotextile surface temperatures on top of snow piles at the BAS (a) and the NBC (b), 2022.

The time series of temperatures at the top geotextile-snow interfaces and different depths in the snow

piles are shown in Figure 9a and 6b at the BAS and the NBC, respectively. While temperatures fluctuated clearly in the upper snow pile due to meteorological conditions at the BAS, the peaks and valleys of the temperature curves in the center snow pile tended to disappear, especially at the 2.45 m depth. The diurnal variations can be detected at the interface and the 0.2 m depth with delayed peak temperatures (5 h of time lag) for TLS1-TLS2, while the 0.45 m depth reaches 13 h of time lag.

Regarding the mean hourly temperature during the same period, temperatures above 0 °C were observed at 13:00 (0.28 °C) and 14:00 (0.24 °C) at the top geotextile-snow interface, indicating surface melting occurred. In contrast, the maximum temperature at a depth of 0.2 m occurred at 18:00 (−3.42 °C). Temperature gradients between different depths (ranging from 0.2 to 2.45 m with 0.25 m intervals) and the geotextile-snow interface showed an increasing trend followed by a decrease for

TLS1-TLS2, with the transition occurring at a depth of 0.7 m with a value of 4.41 °C·m$^{-1}$. The depth of 0.2 m exhibited the smallest temperature gradient (1.71 °C·m$^{-1}$) due to its proximity to the boundary, which facilitates heat and mass transfer. For TLS1-TLS2, the temperature gradient at a depth of 2.45 m was 2.00 °C·m$^{-1}$. From March 8 onwards, a reversal in the temperature gradient's direction was observed, leading to a change in the direction of heat and water vapor flow dominated by the

temperature gradient. The temperatures increased over time, which can be explained by rising air temperature and the thermistor sensors being less distant from the surface or possibly exposed. From the temperature curve (Figure 6a), we can infer that the recorded positions decreased by 2.45 m on approximately April 24.

At the NBC, no diurnal temperature above 0 °C was observed at the geotextile-snow interface and

within the snow pile (at depths ranging from 0.25 to 2.25 m, with a 0.25 m interval). At the interface and the 0.25 m depth, increasing, decreasing, and then increasing trends were detected during the WOcd and WO-WPcd. However, the temperature at the 2.25 m depth did not follow these trends in the upper snow pile but rather continuously decreased. This phenomenon is primarily due to the attenuation and lag of temperature waves by the snow layers. Compared with the BAS, the snow pile at the NBC

experienced more significant temperature gradients between the geotextile-snow interface and several

depths. Additionally, the temperature gradient decreased with depth, with a value of 6.08 °C·m$^{-1}$ at the

2.25 m depth during the pre-competition days (Pre-WOcd) at the NBC. Given the similar initial

conditions of snow piles at the BAS and the NBC, the temperature differences within snow piles are

mainly caused by the meteorological conditions.

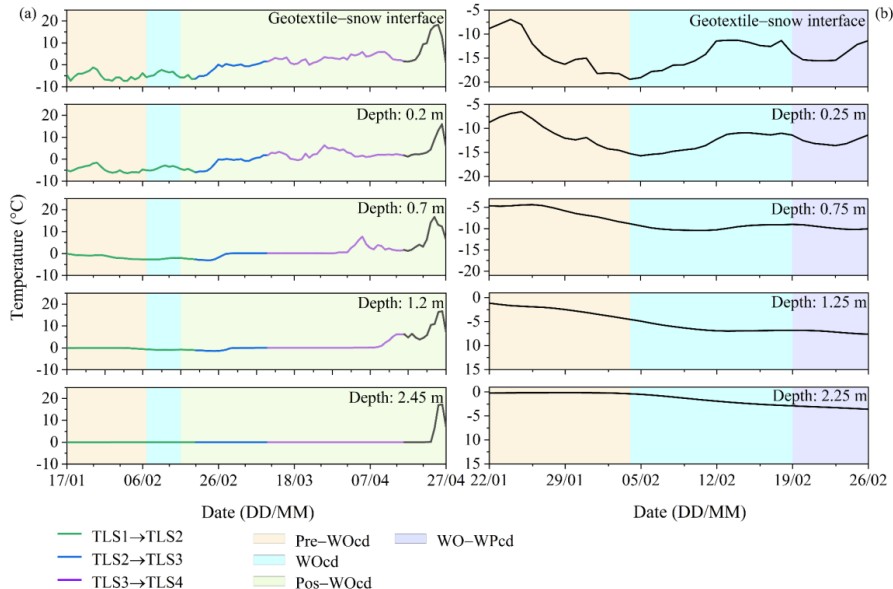

**Figure 6:** Diurnal measured temperatures at the top geotextile-snow interfaces and different depths in snow piles at the BAS (a) and the NBC (b), 2022.

### 4.4   Modeled snow height and snow quality

As mentioned earlier (Sect. 3.4), we focused on modeling the top position evolution of the two snow

piles, where the geotextile-snow interfaces were equipped with sensors and not affected by geotextile

overlaps. This way allowed us to easily parameterize the boundary conditions, irrespective of overlaps,

air layers, and air circulation between the internal and external geotextile layers. Furthermore, it

enabled a meaningful comparison between the modeled external geotextile temperature and the

measured geotextile-snow interface temperature (Sect. 4.5). The measured snow height was derived

from a corresponding 0.05 m resolution grid based on four terrestrial laser scanning surveys. The initial

snow height of this grid was 646.4 cm at the BAS and 572.4 cm at the NBC. We also considered terrain

shadowing caused by surrounding buildings in the model (Bavay and Egger, 2014). Figure 7a



illustrates the modeled temporal evolution of snow height for the two snow piles. At the BAS, the modeled snow height dropped by 268.2 cm between January 17 and April 27. Due to lower air

temperatures, the modeled snow pile at the NBC experienced slight ablation from January 22 to March 4. The modeled downward trend aligns with the measured heat flux at the top geotextile-snow interface at the BAS (Figure 7a and Figure S1). However, the model result slightly overestimated the measured snow height (Figure 7b). The model showed a 3% overestimation of snow height on April 15. This deviation is a result of three effects: (i) wind speed was recorded lower than actual due to the presence

of fences (Figure 1b); (ii) the fouling on the external geotextile surface as mentioned in Sect. 4.2; (iii) inherent limitations of the model itself. On April 24, the modeled snow height dropped by 249.5 cm, in line with the temperature indication (Sect. 4.3). To investigate the differences between covered and uncovered snow piles, we established two schemes in the SNOWPACK model (Figure 7a). Compared to the uncovered scenario, the geotextile offered protection for 45% of the snow height on April 27 at

the BAS. Minor disparities between uncovered and covered snow height are detected in the early stages. However, it would be premature to infer from these results that the geotextile played a negligible role during these periods, as wind erosion on the southwest side of the snow pile was observed at the BAS on February 7, with a maximum difference of 12 cm between covered and uncovered snow surfaces (Figure S2).

Figure 7c and 7d illustrate the distribution of liquid water in the snow piles at the BAS and the NBC. Initially, the mean liquid water content of the geotextile cover was zero until February 25, after which it stabilized at 7–8% at the BAS. Diurnal variations can be detected in the later period. Examination of the flux components revealed that net longwave radiation was the most significant expenditure component in the earlier period, while the latent heat ($-44.38$ W m$^{-2}$) was lower than the net longwave

radiation ($-27.04$ W m$^{-2}$) in the later period. This contrast suggests that the liquid water content of the geotextile contributes to the latent heat and reduces ablation. The liquid water content in the snow layers of both piles showed no significant variation, except for the upper part of the snow piles (Figure 7c and 7d). At the NBC, the mean liquid water content of the geotextile cover was zero, resulting in a small latent heat (Figure 9). In Figure 7e and 7f, we present the modeled snow density for the covered

snow piles at the BAS and the NBC, respectively. The low porosity (mean = 44%) of the initial snow



layers contributed to slight variations in snow pile density at the BAS and the NBC compared to the seasonal snow cover. Assuming the initial snow density within the lower part of the snow pile is the same as at 3 m depth, the snow density increased by 6.4% at the BAS on April 27 and by 2.2% at the NBC on March 4.

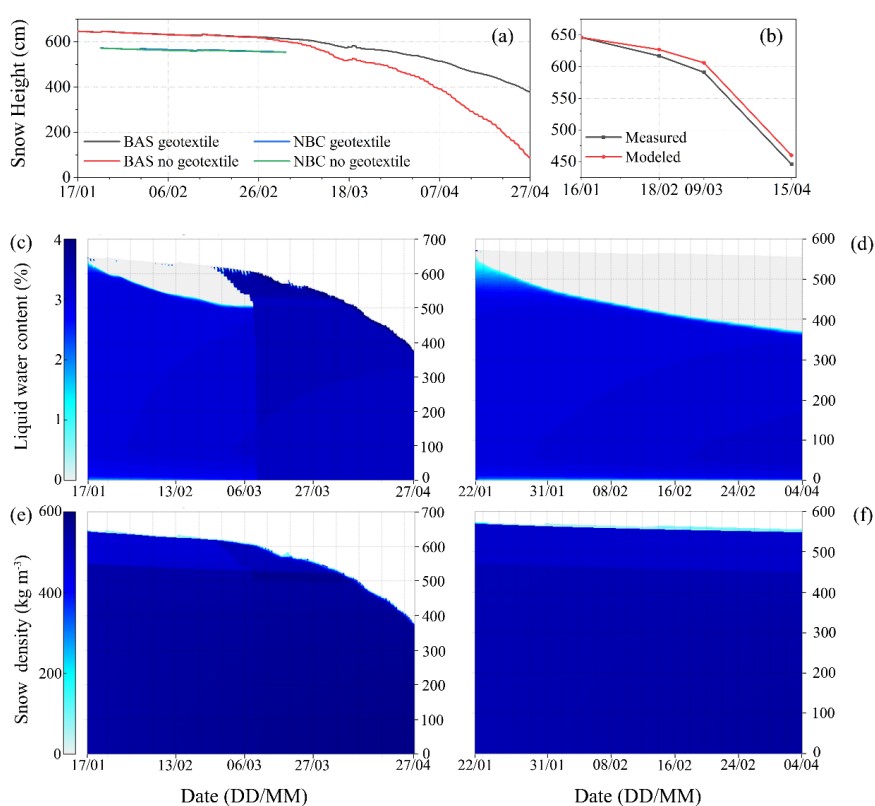

**Figure 7:** Measured and modeled snow piles evolution. (a) Modeled snow height evolution for covered and uncovered at the BAS and the NBC, 2022. Measured snow height for four terrestrial laser scanning surveys and corresponding modeled values at the BAS (b). Modeled liquid water content for covered snow piles at the BAS (c) and the NBC (d). Modeled snow density for covered snow piles at the BAS (e) and the NBC (f).

## 4.5 Modeled top external geotextile surface temperature and energy balance

The external geotextile surface temperature is a significant parameter that serves as Dirichlet boundary conditions. Figure 5a and 5b demonstrate that the model effectively captured the trends in external geotextile surface temperature. Tiny differences between the modeled top external geotextile surface and the measured external surface temperatures on the sides were detected at the BAS due to snowfall.

Early simulations were limited by initial conditions and the model itself, leading to unreliable results. This insight helps explain the differences between the modeled top external geotextile surface and the measured external surface temperatures on the sides at the NBC. Figure 8 presents the modeled top surface (covered and uncovered) and the measured top geotextile-snow interface temperatures at the BAS and the NBC. The relationship between modeled top external geotextile surface and measured top

geotextile-snow interface temperatures can evaluate the model performance. The top external geotextile surface temperature above 0 °C was generally underestimated (mean modeled geotextile surface = 2.8 °C, mean interface = 2.9 °C) at the BAS. This deviation is attributed to an underestimation of convective heat transfer and an overestimation of geotextile reflectivity (Sect. 4.2), SNOWPACK model, of course, cannot be excluded from contributing to this deviation. Contrary to the external

geotextile surface temperature, the additional heat will be utilized for melting snow instead of raising the snow surface temperature when the bare snow surface temperature reaches around 0 °C (Figure 8). Between January 17 and April 27 at the BAS, the modeled mean top external geotextile surface temperature was –1.0 °C, while at the NBC, it was –18.2 °C between January 22 and February 26.

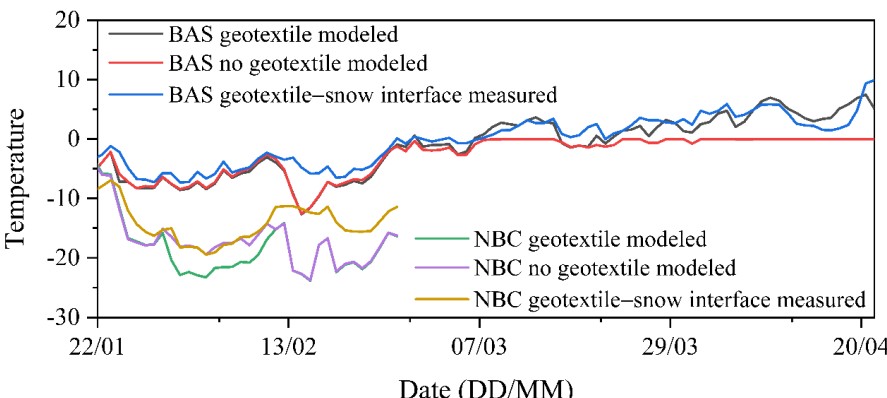

**Figure 8:** Modeled top surface temperatures for covered, uncovered, and measured top geotextile-snow interfaces at the BAS and the NBC, 2022.

The important energy balance analysis (Figure 9) reveals that covered snow piles had lower fluxes than uncovered piles for all components at the BAS. The most significant energy expenditure component at the BAS between January 17 and April 27 was latent heat, indicating that the evaporation of wet

geotextile significantly contributed to protecting the snow pile. In contrast, the net longwave radiation





was the most significant expenditure component at the NBC. The largest incoming component at the BAS was sensible heat, which differs from high-latitude and/or high-altitude regions (Grünewald et al., 2018; Olefs and Lehning, 2010). This contrast highlights the need for improving the thermal insulation capacity of a cover layer in mid-latitude and low-altitude regions. The contribution of heat from rain

was minimal, with 0.01 W m⁻² at the BAS and 0 W m⁻² at the NBC. The energy balance components at the NBC illustrate the conditions for covered and uncovered snow piles under 0 °C air temperature. The mean external geotextile surface temperature (–18.2 °C) was lower than the bare snow surface temperature (–16.9 °C), and both were below the air temperature (–14.5 °C), resulting in greater sensible heat and net longwave radiation for the covered snow pile. Due to higher reflectivity, the net

shortwave radiation for a covered snow pile was lower than that for an uncovered one. The magnitude of incoming energy components was small, and the incoming components were dampened when passing through the geotextile cover. When the external geotextile surface temperature is higher than the bare snow surface temperature (i.e., air temperature is above 0 °C), the conditions are reversed. The sensible heat, latent heat, and net longwave radiation of the covered snow pile are lower than those of

the uncovered pile. As a result, the differences in snow height increase between covered and uncovered snow piles (Figure 7a).

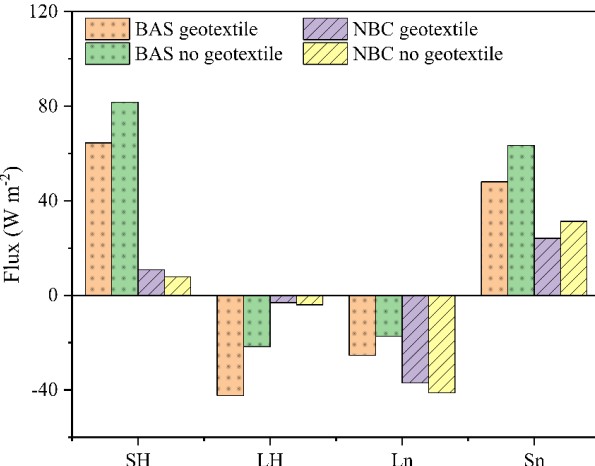

**Figure 9:** Important energy balance components for covered and uncovered at the BAS between January 17 and April 27, 2022 and the NBC between January 22 and March 4, 2022. SH for sensible heat. LH for latent heat. Ln

for net longwave radiation. Sn for net shortwave radiation.



## 5  Discussion

### 5.1  Snowmaking evaluations

Water droplets cool and form artificial snow under low air temperatures and relative humidity conditions. A wet bulb temperature threshold of –2 °C indicates the feasibility of snowmaking (Hartl et al., 2018; Olefs et al., 2010). We calculated wet bulb temperatures for the automatic meteorological stations near the snow storage sites (Ding et al., 2014). At the BAS, daytime wet bulb temperatures were close to or equal to the threshold during the Pre-WOcd, which aligns with the operation of snow machines that snowmaking is unfeasible during the day. However, A time range higher than the threshold existed in WOcd (Figure 10a). 87% of the hours during Pre-WOcd and WOcd were suitable for snowmaking, but some hourly wet bulb temperatures were close to the threshold, leading to reduced snowmaking efficiency and stability and necessitating temporary pauses in snowmaking. The hourly wet bulb temperatures at the NBC (mean = –16.1 °C for Pre-WOcd and WOcd) were significantly lower than those at the BAS (mean = –4.7 °C for Pre-WOcd and WOcd). Snowmaking was feasible at any time before the WPcd at the NBC (Figure 10b). However, there was an apparent increase in wet bulb temperature during WPcd, making snowmaking conditions less favorable. Overall, the snowmaking conditions at the NBC were better than those at the BAS. The daytime wet bulb temperatures were inconducive to snowmaking for Pre-WOcd and WOcd at the BAS, while the snowmaking conditions were even worse during WPcd at the NBC.

We conducted an analysis of wet bulb temperatures for the last three years in Beijing and Chongli based on data from the Mentougou and Chongli national meteorological stations (Figure 10c–10f). Generally, the snowmaking reliability in February 2022 was better than that in 2021 and 2020 in both Beijing and Chongli. Additionally, the snowmaking conditions in March 2022 were better than in 2021 but worse than in 2020 in Chongli. The mean wet bulb temperature for 2021–2022 (December to February) was –3.8 °C in Beijing, which was 0.9 °C colder than 2019–2020 but 0.1 °C warmer than 2020–2021. In February 2022, the wet bulb temperature (mean = –4.2 °C) was much colder compared to 2020 (mean = –1.2 °C) and 2021 (mean = –0.5 °C), contributing to smoother competition processes. In contrast, at Chongli, the overall wet bulb temperature for 2021–2022 (October to March) was colder than the other two periods, with mean values of –6.8 °C for 2019–2020, –7.4 °C for 2020–2021, and –



8.0 °C for 2021–2022.

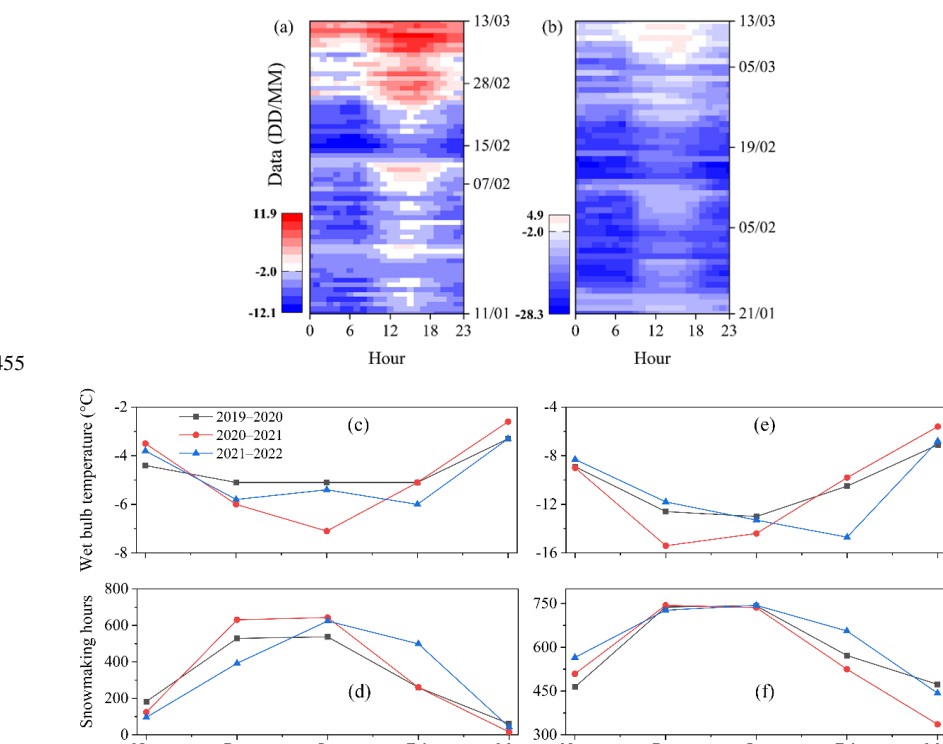

**Figure 10:** Snowmaking reliability. Hourly wet bulb temperature at the BAS (a) from January 11 and March 13, 2022 and at the NBC (b) from January 21 to March 13, 2022. The white colors represent the snowmaking threshold. Statistics of mean wet bulb temperatures and snowmaking hours for Beijing (c, d) and Chongli (e, f) for 2019–2022.

## 5.2 Sensitivity evaluations of snow storage to the main meteorological conditions, cover layer parameters, and snow pile initial conditions

To examine how snow pile height responds to key meteorological conditions, we input various scenarios into the model based on measurements at the BAS from January 17 to March 27. The results reveal that the snow pile height is most sensitive to air temperature (Figure 11a), which is expected due to the limitations of geotextile thickness. Geotextile is an insulation material with excellent thermal parameters, superior to sawdust (thickness = 0.3–0.8 m) which is commonly applied in snow storage (Grünewald et al., 2018; Lintzén and Knutsson, 2018). The thickness is a vital parameter for heat transfer, and thus it affects heat resistance, thermal storage capacity, time lag, and decrement factor of





the cover layer (Ozel, 2013; Wang et al., 2021). However, a very high thickness of tens of centimeters

is not practical due to construction and cost limitations. This characteristic of geotextile makes it unable

to adapt to snow storage under a higher air temperature. Wind speed and incoming shortwave radiation

have less impact on snow pile height (Figure 11b and 11c). The increase in wind speed leads to higher

sensible heat but lower latent heat, while the high shortwave radiation reflectivity reduces its sensitivity

to incoming shortwave radiation. The increased precipitation slightly changed snow pile height (Figure

11d). During the model period, solid precipitation constituted 70% of the total precipitation which

interacts with the geotextile surface through heat exchange, melting, evaporation, and sublimation for

cooling. However, even when the model was run with 15 times the measured precipitation (380 mm) to

exclude the low sensitivity caused by low measured precipitation, the reduction in ablation was

minimal, with an increase of only 15 cm in snow pile height.

The cover layer parameters play a crucial role in the model performance, some of which can be

obtained from literature rather than direct measurements (Lintzén and Knutsson, 2018). In order to

implore the influence of cover layer parameters on the model performance, we selected thermal

conductivity, specific heat capacity, reflectivity, and thickness as variable parameters in the model. The

results demonstrate that the thermal parameters of the geotextile contribute minimally to ablation

mitigation. In contrast, geotextile reflectivity emerges as the most critical parameter in reducing

ablation (Figure 11e). A reflectivity of 0.9 slows down ablation by 21%, while a reflectivity of 0.4

increases it by 35%. It is important to carefully consider the geotextile reflectivity setting in the model.

As previously stated, adding more cover layers can enhance thermal performance. With 18 geotextile

layers (total thickness = 0.072 m), ablation is reduced by 12%. However, this effect is insignificant

compared to the construction difficulty. The 0.4 m thick geotextile layer, main sawdust thickness in

snow storage, reduces ablation significantly by 78%.

Snow pile initial conditions also affect the ablation, and these variations are attributed to grooming

practices and local meteorological conditions. Figure 11g and 11h demonstrate that snow height

increases with density while it decreases with liquid water content. In the density sensitivity

experiments, the liquid water content and grain size were kept constant, that is, the density changed the

volume fractions of ice and air. Conversely, in the liquid water content experiments, the density

remained constant, but the volume fractions of water, air, and ice changed. These results indicate that

the positive impact of grooming practices on snow storage.

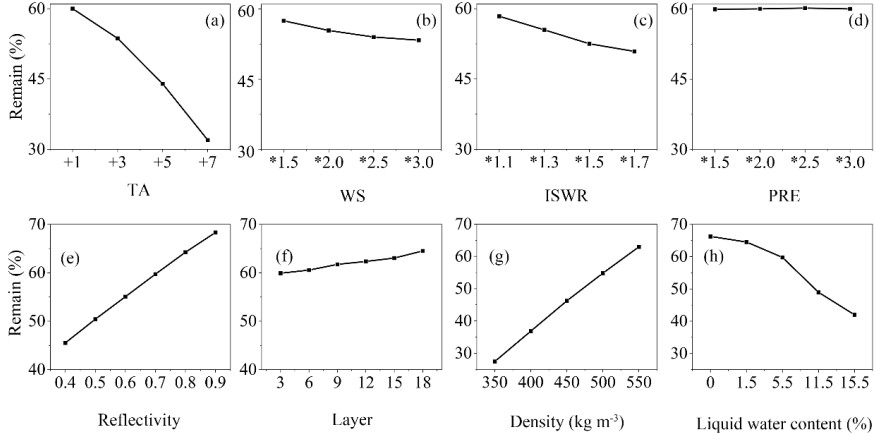

**Figure 11:** Model sensitivity to main meteorological conditions, cover layer parameters, and snow pile initial conditions. (a) Air temperature. (b) Wind speed. (c) Incoming shortwave radiation. (d) Precipitation. (e) Reflectivity. (f) The number of cover layers. (g) Snow density. (h) Snow liquid water content and corresponds to a general classification (Fierz et al., 2009).

### 5.3 Snow storage over the summer season

Preserving snow throughout the summer season is a critical practice in ski industry applications (Grünewald et al., 2018; Ter-Zakaryan et al., 2019; Weiss et al., 2019). Considering the numerous ski resorts and venues in the Beijing and Chongli regions, snow storage becomes indispensable in the context of global warming. Therefore, it is imperative to investigate the evolution of snow piles over the summer season in these areas. The primary meteorological data for the investigation was obtained from the Mentougou and Chongli national meteorological stations between January and October, 2021. To ensure environmental friendliness and ease of shaping, natural thermal insulation materials were considered, with straw being a preferred choice due to its reliable thermal insulation performance and easy availability in northern China. For practical and cost-effective reasons, a combination of two layers of thermal insulation materials was used, consisting of geotextile to reflect incoming shortwave radiation, along with straw chips. Additionally, sawdust was included to examine differences in comparison to straw chips. It is worth noting that in terms of density, straw chips, and sawdust are larger than traditional foam insulation materials (Wang et al., 2021). resulting in their higher thermal





storage capacity.

In the SNOWPACK settings, we determined an initial height of 7 m, a snowpack temperature of 0 ℃, and densities following the snow pile at the BAS. Figure 12 illustrates the modeled top snow height variations with different thicknesses in Beijing and Chongli for January 1 and October 31, 2021. Considering the thin geotextile layer (0.004 m), the second layer thickness is considered the overall cover layer thickness. To maintain at least half of the snow pile height over the summer season in

Beijing, a minimum cover layer thickness of 0.7 m is required, while for Chongli, this value is 0.4 m. Apart from ablation, snow quality, including density, grain size, and liquid water content, is crucial for ski industry applications. The results show minor variations in snow quality over the summer season between cover layer thicknesses of 0.7 to 1.1 m in Beijing and 0.4 to 1.1 m in Chongli. The mean snow densities, grain sizes, and liquid water contents for snow storage were larger than those of the ski track

surface layers (Table 2), indicating that the stored snow can be used for the bottom and middle layers of the ski track. Generally, as the cover layer thickness increased, the increase in snow pile height became less significant. As the cover layer thickness increased, the snow pile height curves in two places kept approving, indicating that the large thickness contributes to snow pile height being incredibly insensitive to meteorological conditions. Additionally, the discrepancies between sawdust and straw

chips can be ignored under larger thicknesses. The above phenomena can be explained from a heat transfer perspective. Heat transfer through the cover layer primarily controls ablation, and it is determined by the heat transfer coefficient, the thermal conditions difference between the internal and external of the cover layer, and the thermal storage capacity. As the thickness increases, the heat transfer coefficient decreases, while the thermal storage capacity increases. In the melt season, heat

transfer from the external surface of the cover layer to the internal surface during the day causes the cover material to warm up and store thermal energy. This process leads to an increase and fluctuation in the internal surface temperature. More heat is stored in the cover layer under large thicknesses, resulting in smaller increases and fluctuations in internal surface temperature, which are also controlled by the specific heat capacity and thermal conductivity. At night, when the air temperature is below the

external surface temperature of the cover layer, the stored thermal energy is released into the atmosphere through convection and radiation. The heat transfer coefficient increases more slowly with





larger thicknesses, with a difference of 0.035 W m$^{-2}$ K$^{-1}$ for 0.4 m and 0.5 m thick sawdust and 0.006 W m$^{-2}$ K$^{-1}$ for 1 m and 1.1 m thick sawdust. Additionally, the thermal storage capacity also shows the same trend with an increase in thickness. The remaining snow height with a sawdust cover was slightly

higher than that of a straw chips cover. The heat transfer coefficient of sawdust (0.07 W m$^{-2}$ K$^{-1}$ for 1 m thickness) is larger than that of straw chips (0.05 W m$^{-2}$ K$^{-1}$), but the volumetric heat capacity of sawdust (242.6 KJ K$^{-1}$ m$^{-3}$) is larger than that of straw chips (60.0 KJ K$^{-1}$ m$^{-3}$). Since the same settings of sawdust and straw chips in the model, except for thermal properties and density, the thermal storage capacity has a more significant impact than the heat transfer coefficient.

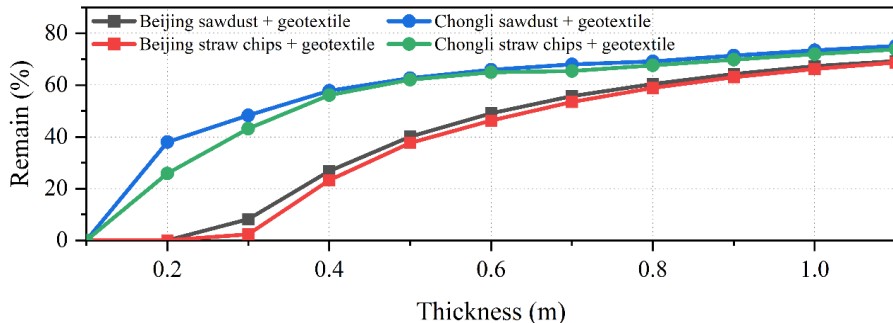


**Figure 12:** The modeled remaining snow pile height varies with different cover thickness at Beijing and Chongli between January and October, 2021. Cover layer: sawdust overlain by geotextile (a reflective layer); straw chips overlain by geotextile (a reflective layer).

**Table 2:** Modeled snow quality for snow storage over the summer season on October 31, 2021 and measured snow
quality at the Wanlong ski resort, China (density and liquid water content for 0.2 m below a track surface, grain size for 0.7 m below a track surface) and a cross-country racing track, Canada (Wagner and Horel, 2011).

| Sites | Density (kg m$^{-3}$) | Grain size (mm) | Liquid water content (%) |
|---|---|---|---|
| Beijing (snow storage) | 605.5 | 2.30 | 3.2 |
| Chongli (snow storage) | 609.6 | 2.29 | 3.2 |
| Wanlong ski resort | 473 | 0.67 | 2.6 |
| A cross-country racing track | 525 | 0.3 | -- |

## 5.4  Limitations

Simulations were not conducted for the snow pile sides, primarily due to various reasons. Firstly, the lateral variations and the influence of radiation and wind make it challenging to obtain accurate
boundary conditions on the snow pile sides (Grünewald et al., 2018). Additionally, the lack of uniformity between the geotextile and snow surface, such as the presence of air layers and overlaps,

further complicates the boundary conditions and makes it difficult to quantify these factors at a high-resolution on the snow pile sides. In a snow storage simulation study, it is a difficult point to obtain high-resolution boundary conditions under a snow pile scale. Moreover, we decided not to explore

other snow cover models in snow storage because of this considerable extra effort and as we wanted to focus on the analysis and evaluation of snow piles variations and test the SNOWPACK model performance at mid-latitude and low-altitude regions, with the aim of providing a tool for feasibility analysis in China.

Another limitation of the study pertains to wind conditions. The location of the meteorological station

being lower than the fences led to an underestimation of turbulent flux. This aspect should be considered in future studies to improve the accuracy of simulations. Naturally, snow quality is a crucial consideration, especially in major winter sports events, encompassing factors like grain size, density, and liquid water content. However, conducting manual snow profiles to assess snow quality would require disturbing the snow piles, leading to a reduction in stored snow and an acceleration of ablation.

Finding alternative methods to assess snow quality without damaging the snow piles would be beneficial.

## 6     Conclusions and perspectives

This study investigated snow storage cases for the Beijing 2022 Winter Olympic and Paralympic Games, with a particular emphasis on two key aspects: ablation and snow quality. The 1-D

SNOWPACK model was first evaluated for snow storage covered with geotextile at the mid-latitude and low-altitude regions. Meteorological data were obtained from automatic stations near the snow storage sites and national meteorological stations, while the geometry and thermal regime of the snow piles were obtained through terrestrial laser scanning surveys and thermal sensors, respectively.

The study draws several important conclusions regarding snow storage in China: a) A significant loss

of 1242.9 m³ of snow (52.7% of the initial volume) occurred between January 16 and April 15 at the BAS. The geotextile effectively protected the snow pile during Pre-WOcd and WOcd at the BAS, but beyond February, the snow storage with geotextile alone became unfeasible due to substantial losses. b) The SNOWPACK model demonstrated its capability to accurately reproduce snow pile evolution in

mid-latitude and low-altitude regions of China. While there was a slight underestimation (7%) in the
top snow pile height drop at the BAS, the model exhibited excellent agreement with measured snow
height, especially considering the sensitivity to deviations of a thin geotextile layer. c) There were
minimal variations in liquid water content and densities of the snow layers at the BAS and the NBC,
except for the upper parts of the snow piles. d) The evaporation of the geotextile surface played a
significant role in protecting the snow pile. Consequently, impermeable coverage schemes for snow
storage are not recommended. e) The study revealed that 0.4 m thick cover layers could protect half of
the snow height in Chongli between January and October, while the thickness needed to be increased to
0.7 m in Beijing. The evolution of snow quality over the summer, with different thicknesses and sites,
exhibited consistent patterns.

While large-scale snow storage requires substantial investment and determination, the specific
quantitative economic effects of snow storage remain unknown. In future studies, it would be beneficial
to consider factors such as transportation, thermal insulation materials, manpower, construction
machinery, snowmaking, and ski resort income in assessing the economic impacts of snow storage.
Additionally, further research on the snow pile sides could provide valuable insights. Given the
importance of snow reliability for ski resorts in southern China, characterized by short snowmaking
hours and high wet bulb temperatures, the study could explore short-term snow storage options for this
region.

**Data availability**

All raw data can be provided by the corresponding authors upon request.

**Author contributions**

DQ, FW, and JR conceived the idea and contributed to the overall framing of the study. XW wrote the
manuscript and collected, processed, and analyzed data. HL contributed to the implementation of the
model. JR and FW revised the manuscript. All authors discussed the results and contributed to editing
the manuscript.





**Competing interests**

The authors declare that they have no conflict of interest.

**Acknowledgments**

This work was supported through the grants from National Key R&D Program of China (2020YFF0304400), Third Comprehensive Scientific Expedition of Xinjiang Uyghur Autonomous Region (2022xjkk0802), and State Key Laboratory of Cryospheric Science (SKLCS-ZZ-2022). The

authors acknowledge the support work of Chunhai Xu, Fanglong Wang, Jiazhen Huang, Jianxin Mou, Xiaoying Yue, and Xin Zhang from the State Key Laboratory of Cryospheric Science. We thank Mathias Bavay from the WSL Institute for Snow and Avalanche Research SLF for the SNOWPACK model guidance.

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
