# Peer review of "Assessing the key concerns in snow storage: A case study for China"

_The Cryosphere, 2023_

## Author Response (AR1)

**Assessing the key concerns in snow storage: A case study for China**

Xing Wang

We would like to thank the two reviewers for giving constructive comments on our paper. We have responded each comment with great care. The original comments of the reviewer are given in italic, and our responses are given directly below in regular.

**Response to Reviewer 1**

Some comments:
*Line 300: It should be Figure 6a and 6b, not 9a.*

**Reply**: We have corrected it.

*Figure 6 shows the temperature in the snow piles at different depths. However, I don't understand the values from BAS. The temperature seems to be above zero degrees in the snow? That is not possible. Can you explain these values.*

**Reply**: It's true that the temperature within the snow cannot exceed zero degrees. However, as the snow pile melts, the sensors will be exposed to the air, leading to recorded temperature exceeding 0 degrees. We have added an explanation at lines 318-320.

"The temperatures increased over time and eventually exceeded 0 °C, which can be explained by rising air temperature and the thermistor sensors being less distant from the surface or possibly exposed."

*Figure 7 - Units on the scale on the right hand side is missing.*

**Reply**: We thank the reviewer for this comment. We now have added units on the right panel.

*In Table 2 - the modeled and measure liquid water content is given. Do you know how the LWC was measured at the Wanlong ski resort?*

**Reply**: Yes, we know. The LWC was measured by the Snow Fork instrument at the Wanlong ski resort. The sensor is a steel fork used as a microwave resonator. We now have added the information in Table 2 caption.

"density and liquid water content were measured by the Snow Fork instrument for 0.2 m below a track surface, grain size was measured by the Anyty 3R-V500IR/UV series optical microscope for 0.7 m below a track surface"

*Line 489-492: Adding more layers will increase the insulation performance. But as you have mentioned, using 18 layers is likely not an alternative at any real snow storage. Also, the sentence "The 0.4 m thick geotextile layer, main saw dust thickness in snow storage, reduces ablation significantly by 78%" could be changed. I suppose you mean that the geotextile+saw dust together will be 0.4m?*

**Reply**: It is not correct that we mean the geotextile+sawdust together will be 0.4 m. We aim to show that sufficiently thick geotextiles can significantly reduce melting, similar to sawdust.

**Response to Reviewer 2**

*Snowfarming and the extended of snow storage is an interesting concern which is addressed by few studies, but on the other hand this snow management practice isn't widely spread among ski resorts. The authors argue it might become more and more critical in order to deal with climate change constraint on snow production (decreasing cold windows favourable to snowmaking) and water availability. The future interest of such practices may be assessed only if the knowledge is sufficient for modelling the snow storage under a changing climate. This article provides original results on the short-term storage of snow during winter, using a material that has been little studied for this purpose. However, the corresponding operational use cases seem rather limited. The references listed show a good understanding of the studied field by the authors and the list is rather exhaustive.*

**Reply:** We appreciate the reviewer's professional insights and have mentioned the future interest of snow storage in "conclusions and perspectives" section.

*The authors deployed a large panel of complementary tools in order to monitor snow piles evolutions. The uses of several means in order to measure the snow related variables and the environmental factors provide a large amount of data and the challenge is to make a comprehensive analysis based on this material. The figures play a critical role to achieve this goal and the ones provided show several limits that need to be addressed before considering publication. Most plots readabilty can be improve. Several suggestions are brought along the article and reported below.*

**Reply:** We express our gratitude to the reviewer for their critical and constructive comments on the figures, which have guided us in making significant improvements to enhance figures clarity and effectiveness.

*The article structure is rather classic but the content isn't always clear enough. Most results are discussed directly and this is particularly confusing when the authors refer to previous or following sub-sections. Later the discussion provides original methods and results. This also has to be improved and made clearer prior considering publication. Particularly, sub-section 5.3 seems to me to go beyond the scope of this article.*

**Reply:** We have improved the content of results and discussions. This paper primarily focuses on assessing the key concerns (ablation and snow properties) in snow storage. Consequently, we removed sub-section 5.1 "snowmaking evaluations". As you mentioned, the application of short-term snow storage in ski resorts is limited. After the application of this short-term snow storage and a significant snow volume loss observed beyond March at the BAS, we have been interested in the ablation and snow properties of long-term snow storage in the same regions where short-term snow storage was applied. So, based on our current understanding, we retained sub-section "Snow storage over the summer season" but incorporated it as a sub-section 4.6 under the results section, clarifying its rational. Furthermore, the density and liquid water content of a snow pile were in accordance with observations from the BAS, and the setup of the SNOWPACK model was identical to that of BAS, except for the covering layer. This

corresponds to the third objective outlined in lines 88-89 of the paper: "to identify significant considerations in snow storage based on the aforementioned investigation and validation of the SNOWPACK model.". Results were confined to ablation and snow properties in sub-section 4.6, with additional information and results being deleted.

*I have some reservations about publishing this article in its current state. I think the reading/understanding could be improved. The main scope is not really clear and the article wants to cover too many topics. I would recommend to focus on the combination of observational and modeling methods for the study of BAS and NBC snow piles. I would also recommend making a clearer distinction between the results and discussion sections, and limiting the amount of new information, methods and results in the discussion section.*

**Reply:** We have removed sub-section 5.1 "snowmaking evaluations". The main scope of this paper is assessing the key concerns (ablation and snow properties) in snow storage. We have combined the results from observations and the SNOWPACK model as comprehensively as possible. Furthermore, we have discussed the SNOWPACK model in conjunction with observational results in sub-section 5.2 "SNOWPACK model and limitations". The discussion section solely comprises sub-sections on "sensitivity evaluations" and "SNOWPACK model and limitations", with limiting the amount of new information, methods and results.

**Additional comments**

*Line 300: Some of the wording is worth checking, for example, snow height vs snow depth or utilize vs use.*

**Reply:** Thanks for your suggestion. We have checked it. Additionally, we have checked the grammar and language.

*Line 31: « low environmental impact and cost-effectiveness » please add reference(s)*

**Reply:** Done.

*Lines 85-88: the links between the different goals is not explicit*

**Reply:** We have revised accordingly.

In summary, the main objectives of this study are: (i) to investigate the evolution of snow piles and the factors influencing them, (ii) to assess the performance of the SNOWPACK model in conjunction with investigation data from China, particularly in mid-latitude and low-altitude regions, and (iii) to identify significant considerations in snow storage based on the aforementioned investigation and validation of the SNOWPACK model.

*Line 99: Ambiguous. What is a snowmaker. Is it the equipment used for the production or the worker who operate such equipment? Please consider systematic rewording*

**Reply:** We have replaced snowmakers with snow fan guns.

*Line 254: « when the sun was in the south, southwest, and west orientations ». How do you distinguish wind and exposure effects?*

**Reply:** We are unable to distinguish the individual effects of wind and exposure. However, it is fortunate that both factors significantly contribute to ablation on the southwest side of the BAS snow pile, while their contribution to ablation on the northeast side is less pronounced. In summary, air circulation, coupled with wind and exposure effects, accelerates the ablation on the southwest side of the BAS snow pile. Conversely, the increased ablation observed on the northeast side is primarily attributed to the effect of air circulation.

*Line 258: « while densification contributed to 141.4 m³. ». It's not clear to me what snow loss mean. Densification reduces snow volume but is not a loss strictly speaking*

**Reply:** You are correct. We have replaced snow loss with snow volume reduction.

*Line 300: 6a instead of 9a?*

**Reply:** Done.

*Lines 303-304: "The diurnal variations can be detected at the interface and the 0.2 m depth with delayed peak temperatures (5 h of time lag) for TLS1-TLS2, while the 0.45 m depth reaches 13 h of time lag." The time lag is difficult to observe on the figures provided.*

**Reply:** You are correct that the time lag is difficult to observe on the figures provided. We have provided a figure in the supplement (Figure 1).

[Figure]

Figure 1: Mean hourly measured temperatures at the top geotextile-snow interface and different depths in the BAS snow pile from TLS1 to TLS2 (The vertical dashed line represents the maximum temperature).

*Line 333: what "snow quality" means is not clear to me. Do you mean snow properties?*

**Reply:** Yes, snow quality means snow properties. We have replaced snow quality with snow properties.

*Line 339: "snow height". I think the correct wording would be "snow depth" or "snow pile height"*

**Reply:** Changed.

*Lines 340-341: "The initial snow height of this grid was 646.4 cm at the BAS and 572.4 cm at the NBC." Maximum grid value?*

**Reply:** they are not the maximum grid values. The values corresponding to the grids where the geotextile-snow interfaces were fitted with sensors and were unaffected by geotextile overlaps at both the BAS and the NBC. This way allowed us to easily parameterize the boundary conditions, irrespective of overlaps, air layers, and air circulation between the internal and external geotextile layers. Furthermore, it enabled a meaningful comparison between the modeled external geotextile temperature and the measured geotextile-snow interface temperature.

*Line 347: Figure S1?*

**Reply:** Figure S1 in the supplement.

*Line 348: "The model showed a 3% overestimation of snow height on April 15". It*

*seems the overestimation is almost stable from February 2 to April 15. As the percentage is relative to the total amount of snow, please consider adding some information about the absolute value and its evolution between the different measurement dates. The first period seems to be critical.*

**Reply:** Done, we have analyzed the snow pile height drops rather than the snow pile height itself, and have added the differences between modeled and measured snow pile height drops across three periods.

However, the model slightly underestimated the measured snow pile height drop on April 15 (Fig. 7b). The deviations between modeled and measured snow pile height drops were -10 cm (-34%), -5 cm (-19%), and 1 cm (1%) for the periods January 16 to February 18, February 18 to March 9, and March 9 to April 15, respectively. As air temperatures rose (Fig. 1), the gap between modeled and measured snow pile height drops diminished, eventually trending towards overestimation, a deviation consistent with previous studies (Grünewald et al., 2018; Olefs and Lehning, 2010). Interestingly, at lower air temperatures, the model underestimated the measured snow pile height drops for the first (-1.9 °C) and second (4.6 °C) periods.

*Lines 349-350: "wind speed was recorded lower than actual due to the presence of fences (Figure 1b)". I don't understand what you mean and why you didn't record the actual wind speed if the fences were actually there.*

**Reply:** The BAS weather station was positioned lower than the fences, while the top of the snow pile reached a height greater than the fences. Consequently, due to the shielding effect of the fences, the wind speed recorded at the station was lower than that which could be experienced at the top of the snow pile. Regarding why you didn't record the actual wind speed if the fences were actually there, the BAS administration dictated that the weather station had to be situated within the fences perimeter and its height could not surpass that of the fences. We have improved the sentence in lines 358-360.

the wind speed was recorded lower than actual due to the weather station being positioned lower than the fences, thereby subject to the shielding effect of the fences (Fig. 1c).

*Line 352: "in line with the temperature indication (Sect. 4.3)". What do you mean by "in line"?*

**Reply:** We meant that the modeled snow pile height agrees well with the temperature indication. In Fig 6a, we can infer that the recorded location decreased by 245 cm on approximately April 24, deduced from the temperature rise to 0 °C, which closely matches the modeled snow pile height of 249.5 cm. We have modified the sentence to make the expression clear in lines 362-364.

On April 24, the modeled snow pile height dropped by 249.5 cm, consistent with temperature indicators at a depth of 245 cm within the snow pile, indicating a drop in snow pile height of approximately 245 cm (Fig. 6a).

*Line 362: "after which it stabilized at 7–8% at the BAS". How that can be read with a colorbar scale from 0 to 4?*

**Reply:** We appreciate this important comment. Upon reviewing the liquid water content (LWC) data output from the SNOWPACK model and the LWC figures produced by niViz (a visualization tool for the SNOWPACK model), we have identified a possible reason for the colorbar scale being limited to a range of 0% to 4%. This limitation is attributed to the significant difference in LWC between geotextiles and snow, where in some cases, the LWC of geotextiles can exceed 40%. Due to an uncovered interval of LWC between geotextiles and snow, the colorbar display is consequently limited to the range within snow, that is 0% to 4%. To avoid further controversy, we have removed the LWC figures and have made improvements in representing the LWC of geotextiles in lines 372-374.

Initially, the modeled mean liquid water content of the geotextile cover was zero until February 25, after which it reached a mean of 5.5% at the BAS. Diurnal variations can be detected in the later period, peaking at 41%.

*Line 382: "Figure 5a and 5b demonstrate". I don't think a figure demonstrates anything. It shows something.*

**Reply:** We think Figure 5 demonstrates and shows something. 1. The measured and modeled external geotextile surface temperatures were lower than the air temperatures at the BAS and the NBC. 2. The external geotextile surface temperatures on the southwest side were found to be more sensitive to meteorological conditions than the northeast side at the BAS. 3. The SNOWPACK model effectively captured the trends in external geotextile surface temperatures. Furthermore, we have improved Figure 5 in accordance with the subsequent comments.

*Lines 408-409: "This contrast highlights the need for improving the thermal insulation capacity of a cover layer in mid-latitude and low-altitude regions". Actually these are the areas where snow storage seems most critical because in the other case (high latitude and/or high altitude), the capacity to produce snow is also better.*

**Reply:** We agree with the reviewer's perspective. We have now added this point in lines 418- 420.

Furthermore, considering the enhanced snowmaking capabilities and more favorable snowfall conditions in high-latitude and/or high-altitude regions, the importance of snow storage escalates in mid-latitude and low-altitude regions.

*Fig1: where are the fences?*

**Reply:** We aimed to display the entire BAS snow pile; hence, the fence is indistinct in Figure 1a. We have now added a figure of the fences in Figure 1.

*Fig2: The plots should use similar scales in order to be directly comparable.*

**Reply:** Done.

*Fig3: e, f, g, h instead of c, f, g, h? The classes should be the same for i, j and k*

**Reply:** Done.

*Fig4: One more time, I'm disturbed by the incomparability of scales between plots. Here it less disturbing than previously because there is no side-to-side reading/comparison to be done. However, it is required at least to have consistent ranges between the two scales (same lengths) in order to be able to compare the variations amplitude. Please specify which variable is represented (raw observations? daily means? Hourly?) in the caption.*

**Reply:** We assume you were referring to Fig 5, as Fig 4 does not exhibit any variations amplitude. We have accordingly improved Fig 5 based on the reviewer's suggestions.

*Fig6: 0 is a critical temperature threshold. Please consider adding a dashed line, at least for this value but maybe also for the others values labelled along the y axis, maybe using different colors for 0 and the other values. One more time, scales are messy: they don't allow comparison between left and right subplots and moreover for the right-side subplots, the scale change between the first subplot and the last one*

**Reply:** We are grateful for the constructive comments. We have implemented different colors for 0 and other values and addressed all scales.

*Fig7: 7a and 7b: The use of the same colors for geotextile/no geotextile and measured/modeled is confusing. 7c to 7f: Y scales should be the same and the label and units for the right axes added*

**Reply:** We have improved the figure in accordance with the reviewer's suggestions. The modeled line in Fig 7b represents a segment of the BAS Geotextile line in Figure 7a, hence the black color for both lines.

*Fig8: Please consider adding a dashed line for the 0 value at least*

**Reply:** Done.

Grünewald, T., Wolfsperger, F. and Lehning, M., 2018. Snow farming: conserving snow over the summer season. The Cryosphere, 12(1): 385-400, http://dx.doi.org/10.5194/tc-12-385-2018, 2018.

Olefs, M. and Lehning, M., 2010. Textile protection of snow and ice: Measured and simulated effects on the energy and mass balance. Cold regions science and technology, 62(2-3): 126-141, http://dx.doi.org/10.1016/j.coldregions.2010.03.011.